# Development and Sustainability of Rural Economy of Pakistan through Local Community Support for CPEC

Inam Ullah Khalil [1], Sehresh Hena [1], Usman Ghani [2,3], Raza Ullah [4], Inayatullah Jan [5], Abdul Rauf [6], Abdul Rehman [7], Azhar Abbas [4,*] and Luan Jingdong [1,*]

1 College of Economics and Management, Anhui Agricultural University, Hefei 230036, China; inamkhalil.regi@yahoo.com (I.U.K.); sehresh.hena@ahau.edu.cn (S.H.)
2 College of Education, Zhejiang University, Hangzhou 310000, China; gusman@mail.ustc.edu.cn
3 Department of Business Administration, Iqra University, Karachi 75500, Pakistan
4 Institute of Agricultural and Resource Economics, University of Agriculture, Faisalabad 38000, Pakistan; raza_khalil@yahoo.com
5 Institute of Development Studies (IDS), The University of Agriculture, Peshawar 25130, Pakistan; inayat43@aup.edu.pk
6 School of Management Science and Engineering, Nanjing University of Information and Technology, Nanjing 210044, China; abdulrauf@seu.edu.cn
7 College of Economics and Management, Henan Agricultural University, Zhengzhou 450002, China; abdrehman@henau.edu.cn
* Correspondence: azhar.abbas@uaf.edu.pk (A.A.); luanjingdong@ahau.edu.cn (L.J.); Tel.: +86-13505610858 (L.J.)

**Abstract:** This paper investigates the local community support for the China Pakistan Economic Corridor (CPEC) using the lens of social exchange theory. The study examines the direct effect of social, economic, cultural, and environmental factors on the local community support for CPEC projects, and the resultant impacts on the development and sustainability of the rural economy of Pakistan. The study also explores the moderation effect of media influence on shaping positive perceptions of CPEC among the local populace. The local communities at the CPEC route which are affected or can be affected by the project were targeted for data collection using a structured questionnaire. The collected valid data (N = 259) were thoroughly analyzed by obtaining reliability and validity statistics, a correlation matrix, multiple regression, moderation analysis, and hypotheses testing. Our results substantiate that the local community support for the CPEC project is heavily dependent on social, economic, cultural, and environmental factors and that there is a positive influence of media impact as an opinion-maker in the local community regarding the CPEC. The CPEC is expected to develop the rural economy, particularly through improvements in agriculture and allied activities, thereby providing livelihoods and income-generating opportunities to the rural masses. The article is important for regulators, the CPEC authority, government bodies, and the relevant community.

**Keywords:** rural development; sustainability; CPEC; media influence; agriculture

## 1. Introduction

The local populace in Pakistan has different views about the projects launched through the China Pakistan Economic Corridor (CPEC). The local areas' infrastructure exhibits a picture of the socio-economic fabric, and such facilities represent a sign of the country's development. Besides other socio-economic benefits, the project mainly contributes to the promotion of education, health, and energy [1]. CPEC is an offshoot of the Chinese President's ambitious idea, known as the one belt one road (OBOR) initiatives, that include multiple sub-projects with neighboring countries [2,3]. The friendship between China and Pakistan is considered crucial for both countries, as both countries have mutual political and economic interests for the development of the region [4,5]. The China OBOR initiatives

consist of several macro-projects that are spread over most of the Asian and African countries [6]. CPEC was launched by the Chinese government for the socio-economic and infrastructural upgrade of Pakistan [4]. The project is rightly called a game-changer in Pakistan's history and is also crucial for the Chinese side to access a huge market through Pakistan's Gwadar port [7].

In more specific terms, the CPEC is a part of the Belt and Road Initiatives (BRI) connecting from the Xinjiang province of China to Gwadar (Baluchistan province) Pakistan, with an approximate length of 3000 km. As a main pillar of BRI, the project has geostrategic, economic, and environmental implications for both countries. The project has many micro- and macro-level plans for railways, road and transport, fiber and optics, energy, and other pipelines, as well as various industrial parks and zones [8]. The Chinese BRI is one of the most promising projects that is intended to transform all the connected countries along the proposed route to a new height of development [9]. Several studies focused on the impact of the project on the development of the involved countries, but they ignored the local perceptions that can spur the success of the project. Some studies, like those of Callahan [10], Clarke [11] and Fanell [12], looked at the strategic goals of the Chinese government in BRI and found that the project enhances the regional power and connectivity issues. Similarly, Zhang [13] and Shariatinia and Azizi [14] highlighted the cost and economic efficiencies of the BRI that stated that it could encourage trade and economic development in the region.

The positive perceptions of local residents do matter in the success of any project, and this study highlights such perceptions with a comprehensive model. The previous studies, focusing on the local residents' perceptions, ignored the media propagation, which has a huge impact on the minds of the populace. The media presentation has a major role in shaping community perceptions towards the success and/or failure of international projects [15], but there is no comprehensive study available in Pakistan highlighting the impact of media in supporting the CPEC. The national newspapers, magazines, and other print, as well as electronic media, make an especially huge contribution in favor of or against such mega-projects [16], and it is imperative to understand the influence of media in order to deeply comprehend the success factors of CPEC. The CPEC is also heavily covered by international media highlighting its potential strengths and weaknesses [17], but there is sparse literature investigating the local media effects on community perceptions and positive opinions.

The main purpose of this study is to inspect the local community perceptions of the CPEC project. Positive perceptions of the project may boost the effectiveness of the project. More precisely, this study has some novel contributions to make, probing the societal impacts of the local populace on the CPEC project. Similarly, this study evaluates the impact of CPEC on the culture of local residents. Past studies identified that cultural values have a key role in adopting or modifying users' perceptions [18]. The cultural orientations of the local community may be aligned with the development of the project, and therefore this needs to be examined. The environmental concerns of the local residents are also taken into consideration, and thus this contributes to environmental research in the context of international projects. The more focused-on area in the project is the economic well-being of the local community, and the study covers what the local populace thinks of such benefits in terms of their current and future betterment. The economic activities of the local populace revolve around the agriculture sector. Developing and reshaping agricultural production and distribution strategies under the CPEC will be of great value to the local communities in terms of benefits derived from the CPEC project. This study also assesses the moderation role of media influence. The media's role is indispensable for the success of any international project [15], hence, the study suggests that the media influence should be incorporated to sketch a thorough picture of the project's future. This study investigates the media influence which it deems as responsible for shaping up public opinion about CPEC. The influence of media coverage has rarely been researched in the context of CPEC, and thus this study leads to filling the gap in research by evaluating the community perceptions founded by local media.

## 2. Literature Review

### 2.1. BRI and CPEC

The BRI and CPEC are Chinese flagship projects gaining around-the-clock attention from researchers in different fields. BRI covers many of the Asian countries which are considered the hub of the global financial market, and the project is intended to extend to Africa and European markets. The South Asian Association for Regional Cooperation (SAARC) members are all included in the BRI umbrella, except India, which shows some reservations [19]. The BRI includes many corridors, but the central and functioning project is the CPEC, which connects the Pakistani province Baluchistan with the Western province, Xinjian of China, through roads, railways, and pipelines costing around 62 billion USD [20]. The project gained momentum when the Chinese president Xi Jinping and Pakistani prime minister Nawaz Sharif (along with provincial chiefs) met in the "road and belt summit" in 2017 in Beijing [21].

CPEC is a multi-sectoral project at the Western coast of Gwadar, Baluchistan that links all the provinces of Pakistan with roads, pipelines, railway tracks, energy infrastructure and industrial projects, as well as with other sub-projects in communication, health, education, and transportation [22]. The project will also help Pakistan to develop its energy resources along the CPEC route [23], and it will be extended to the other neighboring countries like Afghanistan and other central Asian countries. Three routes are proposed for CPEC in Pakistan: The eastern route will pass mainly through the provinces of Punjab and Sindh [24], while the second route, called the central route, encompasses Khyber Pakhtunkhwa and some underdeveloped areas in Punjab and the Sindh province. The third western route travels rural areas of Baluchistan with Khyber Pakhtunkhwa [25].

The CPEC project is the byproduct of the mutual trust and commonality of thinking between the two countries. During hard times, the Chinese government extended support to the Pakistani government, especially when there was pressure on Pakistan from the international community after nuclear testing and other missile programs [26]. In the late decades of the twentieth century, China remained neutral on many international fronts and that helped it to improve its economy and technology. The twenty-first century, especially the second decade, proved China to be an international player and linked many countries to the economic and technological net, developing friendly and peaceful relations [27].

### 2.2. Theoretical Background and Hypotheses Development

The study is theoretically supported by the social exchange theory (SET) [28]. Playing a key role in motivating primary orientations, the SET postulates that social interactions are the product of mutual benefits, where it is assumed that the benefits are more than the costs of the project. The socio-economic well-being is a two-way process, where both sides (parties) expect more benefits with reduced costs. SET assumes that human interactions are fostered in the case of rewards and are reduced in non-rewarding situations. If the local community considers that the CPEC is economically, socially, culturally, and environmentally beneficial for them, they will welcome it, and vice versa. The higher the benefits of the CPEC project, the higher the support for the project by local inhabitants [29]. The SET has been applied by many researchers in different contexts but has scant applications to understanding societal acceptance and support for international projects. This study contributes to the SEB-based literature to evaluate social interactions with the resultant gains that the local community will acquire from the CPEC project.

#### 2.2.1. Economic Impact

The CPEC project is believed to be economically beneficial for the local residents [30]. Economic impact refers to economic benefits such as earning opportunities, employment, trade, and other related financial benefits that the project offers to local inhabitants. The perceptions of the local community regarding CPEC may be in favor of or against the proposed benefits. Some local residents think that the project will make their living standard higher [31], while others consider the project pessimistically and think that it will

have negative impacts on their community and local industry. The CPEC is a multi-sectoral mega-project with billions of dollars of investments in roads/motorway constructions, railway tracks and system, telecommunications, infrastructural development, and other means of transportation that are expected to produce hundreds of thousands of jobs, various markets for goods and services, logistics and supply chains of valuable products that will all be available to the community with low effort and cost. It was also emphasized that local residents will get their due benefits, with the provision of sophisticated agricultural products, and growth of small and medium-sized businesses like workshops, hotels, and restaurants, storages, petrol pumps and parking facilities. [32]. Moreover, the infrastructure development has a strong association with agricultural output. Investment in infrastructure is directly translated into increased agricultural output [33–42].

The unemployment problem is Pakistan is worsening day by day, due to the growth in population along with many untapped resource, which leads to poverty, hunger, and social exclusion [3]. In these critical conditions, the CPEC might prove to be a blessing for employment generation, and sustainable, and exclusive economic growth in Pakistan. Past studies highlighted that the CPEC will upsurge revenues and help to create economic zones, and cause tourism development which will not only strengthen the local community, but will also change the local inhabitants' mind towards its success [43]. The economic effects would lead to more acceptance for any international project, as it is rightly considered to be the main reason for justification and support. In this context, the current study proposes that the higher the economic impact, the higher the local inhabitants' support for the CPEC project will be. This is, therefore, hypothesized [31]. The development of infrastructure and the resultant growth in agricultural output will also generate employment opportunities in the rural areas, and thereby help in developing the rural economy.

**Hypothesis 1 (H1).** *The perceived economic impact of local residents has a positive effect on the local residents' support for the CPEC projects.*

### 2.2.2. Social Impact

Social impact refers to the influence of CPEC on social interactions and social cohesion, i.e., the social welfare of any society, which is usually indicated by education, health, housing, and other facilities which constitute the needs of common men in any society [3]. Family life, schools, social gatherings, social institutions, and all other social contracts are expected to be enhanced with the CPEC development. Rafiq and Weiwei described the social trusts which comprise behavior, thinking, and the promotion of living standards, which provide a progressive perspective of the CPEC project in Pakistan [44].

Past studies identified many positive impacts of the CPEC on society, such as access to markets, educational networks and social interactions, bridging social breaks and discrimination, and, more importantly, solutions to social exclusion [22]. The overall society, especially the rural areas, will turn into the nucleus of business and social engagements due to the economic and trade influx from China. Such developments are a manifestation of social prosperity, the betterment of healthcare, and the development of smart cities and towns [30]. Socially, the people-to-people exchanges, cultural events, and national days of celebration of both the nations might inaugurate a new era of friendship and relations [2,30,45]. The rural areas will cultivate new thinking due to the CPEC projects, which will move the local populace towards support of the project. This study assumes that the local inhabitants will extend their support to the CPEC projects, believing that the project is a means of social uplift.

**Hypothesis 2 (H2).** *The perceived social impact of local residents has a positive effect on the local residents' support for the CPEC projects.*

### 2.2.3. Cultural Impact

The life standard of the host community is heavily dependent on the social and cultural changes that are introduced by foreign projects [2]. The local culture embraces new, positive changes with infrastructural developments and with the expansion of tourist destinations. The CPEC route might facilitate social interactions between different communities that, in turn, will create new cultural exchanges and events [30]. The new cultural developments from CPEC can be anticipated by increasing close contacts between different families, friends, and peers [29], which is why the perceptions of local residents help to devise policies that minimize the negative impact of the CPEC [22]. International projects (such as CPEC) have a significant impact on the culture of the host community. Such projects give rise to the excavation and exploration of new historical sites, help archeological activities and studies, provide opportunities for cultural exchange, and develop museums and libraries for cultural artifacts. Such developments enhance recreational and entertainment facilities that offer support to the CPEC in the minds of the local community. Some authors pointed out that new cultural values may arise, which might not match the local culture in terms of traditional and religious values, as some locals may think that the CPEC is a threat and a means of Chinese cultural hegemony. The CPEC road and transportation routes may help expedite the provision of goods and services as well as serve as a means of exchanging social customs, and the dissemination of languages and beliefs, among many other things.

However, based on the rationale of SET, the local community will note the benefits of such projects. Based on the previous studies, we believe that the CPEC provides more advantages to the local community, therefore, the perception of it will be positive and the local populations will support the project.

**Hypothesis 3 (H3).** *The perceived cultural impact of local residents has a positive effect on local residents' support for the CPEC projects.*

### 2.2.4. Environmental Impact

Today, environmental implications have been given more importance and are prioritized over many other issues. Previous studies signify that developmental projects increase the environmental issues of any country [29,46]. The CPEC, with many energy and power generation plans involving the use of coal, may deteriorate the environment of Pakistan, especially in those areas where the plans are to be initiated [47]. The local residents' perceptions regarding environmental issues are quite significant for the successful implementation of the CPEC project. The multiple construction activities need numerous energy resources like coal, gas, steel, rubber, iron, and other materials whose excessive use is usually considered to be detrimental to the environment [48].

The transportation and roads project of the CPEC has been found to affect the environment negatively [22], because of the excessive congestion of traffic, overcrowded roads and hotels, noise and pollution emissions of the vehicles, as well as the degradation of air quality, creation of garbage and filth, and the cutting of different trees or deforestation, which may harm the natural environment and beauty [49]. The energy and power projects under the CPEC also tend to negatively impact the environment. Approximately 19 projects to be initiated under the CPEC consist of wind-based, hydel, solar, and coal-based multiplans that will affect the environment. The CPEC also includes some environment-friendly projects like the construction of parks, clean water schemes, renewable energy schemes, picnic and entertainment spots, and pollution control mechanisms that can reduce the degradation of the environment [29].

In terms of the environment, the local community might perceive that the benefits of the CPEC projects are fewer than the costs. Such perceptions will reduce local support for the projects [22]. The local community will welcome the new developments under the CPEC if they think that there is a parallel system to save the environment, and the sustainability of the projects is maintained. Keeping the previous literature in mind, we

assume that the local inhabitants will have negative perceptions about the CPEC projects, and the following is proposed:

**Hypothesis 4 (H4).** *The perceived environmental impact of local residents has a positive effect on the local residents' support for the CPEC projects.*

### 2.2.5. Total Impact

The impact of the CPEC projects on the community has been thoroughly examined in the literature. The CPEC is a flagship and mega project of the Chinese government with Pakistan. The influence of the CPEC on the host country's residents has different dimensions. The total impact is the sum of all the various impacts that the CPEC has on the local inhabitants [22]. Some effects are adverse, while some are very productive and positive for the local areas. All such interactions and consequences lead to the acceptance or rejection of the projects. The main theme and findings of previous studies identify that the gains might be greater than the costs of the projects [50]. Looking at the past patterns, this study proposes that the total positive impact of the local community will have positive effects on the acceptance and support of the project.

**Hypothesis 5 (H5).** *The perceived total impact has a positive effect on the local residents' support for CPEC projects.*

### 2.2.6. Perceived Media Impact

The media tends to influence every individual in society. The latest developments in the media, especially social media, which is interactive in nature, have a huge impact on opinion-making and helps to shape community perceptions. Social media engages people in commenting on other posts, sharing news, and liking and disliking of posts, and all that leads to specific perceptions [51]. The central role of media in the promotion of CPEC was investigated by Asif and Ling [17], and it was found that some of the reporting is credible, while some media outlets speak against CPEC, thereby creating confusion among the local community in Pakistan. The media intentions and the contents of their coverage do influence individuals to either support or be against CPEC.

Due to the persistent growth of the Chinese economy and the success of their international projects, the CPEC is always debated in the national/international media and reporting [52]; thus, the debates continuously frame the perceptions of local residents, Haider and Waqar recently stated that the level of understanding of common men is still unclear, and that the print media offers a narrative about the CPEC which needs to be assessed [53]. The local print media, which is mainly in the local language of Urdu, has a very strong impact on ordinary citizens, especially in rural under-developed areas of Pakistan. Investigating the impact of the Urdu newspaper on the CPEC project, Nazir [54] found that the Urdu newspaper offers a positive portrayal of CPEC, especially in terms of security in the region, peach initiatives, China–Pakistan relations, and the development of infrastructure. Such newspapers are read throughout the country daily, which gives a projection of the CPEC to local residents. The writers also highlighted some areas of politics where the portrayal of CPEC is not as positive and raises some critical questions in local inhabitants' minds.

The persistence and increasing media influence, sometimes called mediatization, is believed to overwhelm the thinking and perceptions of viewers/readers. The "prosumers" in media tend to be influential and lead public opinion [55], and such persuasions may be used as a weapon to foster support or spread negative news about a certain project. The core idea is that local people can be persuaded or dissuaded by media outlets, and this has been found in past studies [15,56]. This paper assumes that the media channels can moderate relationships between the total impact of CPEC and support for the CPEC project as depicted in Figure 1.

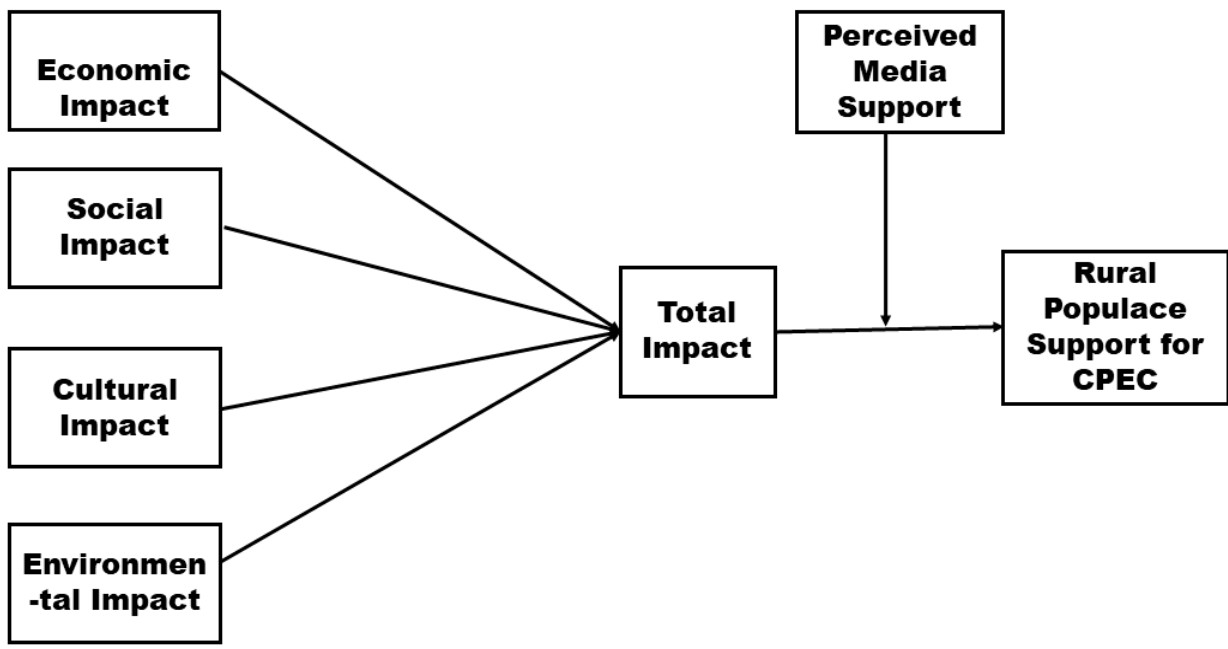

**Figure 1.** Proposed Research Method.

**Hypothesis 6 (H6).** *Perceived media influence moderates the relationship between total impact and the rural residents' support for the CPEC projects.*

2.2.7. Proposed Model of the Study

Figure 1 is proposed research method.

## 3. Methodology

### 3.1. Sampling and Data Collection

To accomplish the objective of the study, this study was conducted in three districts: Punjab, Sindh, Baluchistan, and Khyber Pakhtunkhwa provinces, that exist on the CPEC route (see Figure 2). The objective of this study is to discover the local residents' perception of and support for CPEC development. A total of 1500 questionnaires were administered to the local people residing on the CPEC route in the above four provinces. The respondents included agricultural producers and other stakeholders involved in agriculture and allied activities. The study used convenience sampling, as was used in many previous studies [22], and this is suitable in the context of understanding residents' perceptions.

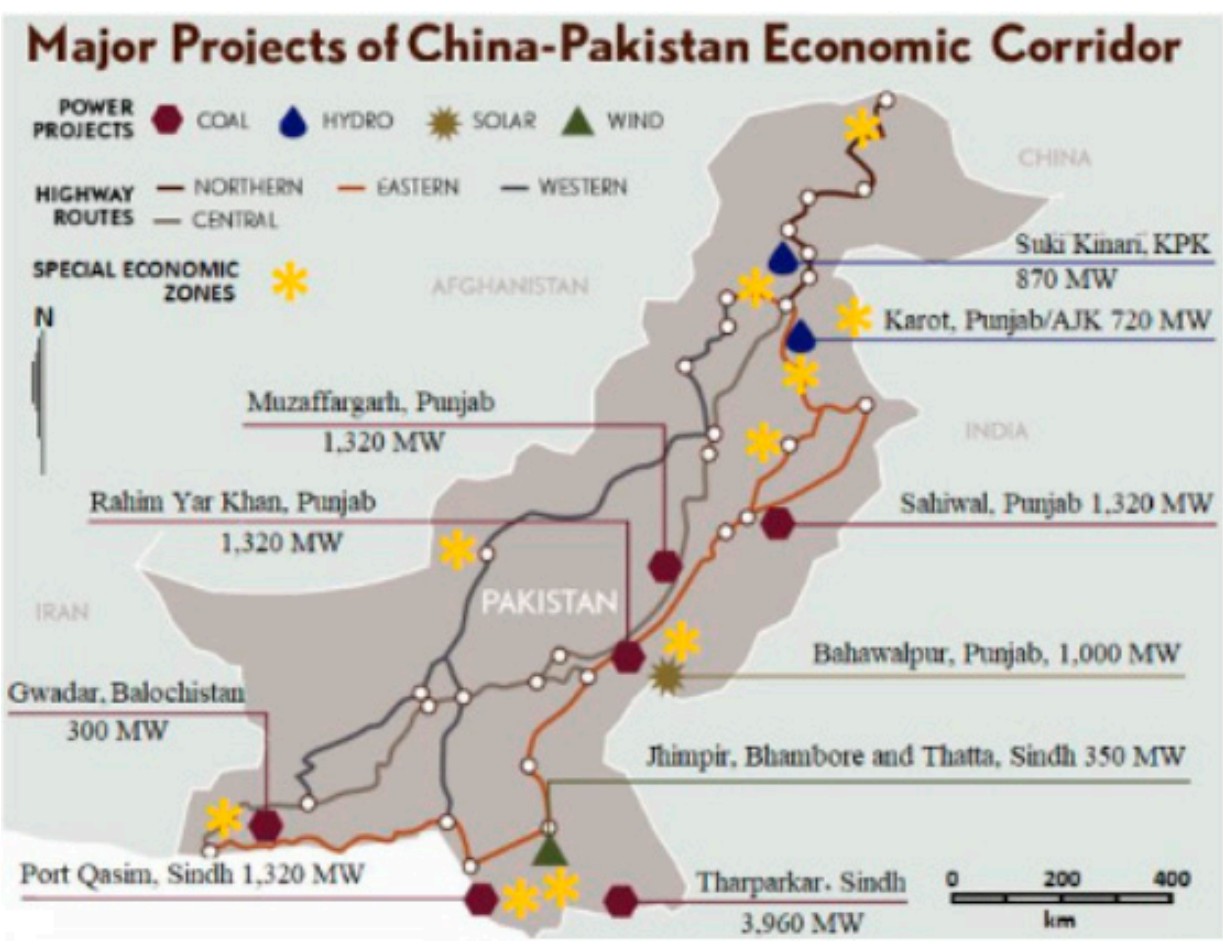

**Figure 2.** Map indicating Major Projects of CPEC in Pakistan (source, Farooqui and Atab, 2018).

### 3.2. Sample Size Calculation

Several methods exist for selecting a suitable sample size. This study applies the estimations of Gorsuch's rule [57], which states that a suitable sample size should be more than five times the total number of questions. The questionnaire of this study is composed of a total of 25 questions, and therefore the resulting sample size should be at least 125 (25 × 5). We also took the help of the Mike Petres [58] methodology for determining sample size and, to that end, the sample size of our study is quite acceptable. In our case, the sample size is 259, which is fairly valid and also has a sound backing from previous studies. All the participants took part in the survey in their free time, and with their free consent. They were briefed that the survey data will be used confidentially, and they will be given the results of the study once it is published.

### 3.3. Measurements

The measurement of our variables was adopted from previous studies used in the context of support for international projects. The measurements and conceptualizations by McGehee and Andereck [57], Yoon [59] and Jurowski [60] were relied upon for this study. The scale used for the current study was a five-point Likert scale (1 = strongly oppose and 5 = strongly support). The five-point scale is commonly used and is valid for such studies [32,61]. All the questions in the instrument were thoroughly checked for language clarity, suitability and understandability. The questionnaire was also examined by the researchers and professors to ensure that it clearly met the objectives.

### 3.4. Data Analysis

The study conducted a focus group for checking the content and face validity of the instrument. Some of the items were deleted as per the suggestions of the focus group, thus adding to the reliability and understandability of the instrument. A quantitative data analysis approach through SPSS was applied to data collected from the residents of seven districts lying on the CPEC route in Baluchistan, Punjab, and KP, Pakistan. For the sake of analysis, first of all, data-cleansing techniques were applied. Exploratory Factor Analysis (EFA) and Confirmatory Factor Analysis (CFA) were used to test the validity and reliability of the questionnaire, and form make factors based on their correlations, and the regression analysis was conducted to test the relationship between the latent and observed variables.

### 4. Results

Table 1 gives details about the respondents in terms of gender, age, education, and province (domicile), which indicate the participation of the respondents across all the four provinces located on the CPEC route.

**Table 1.** Demographic information.

| Demographic Characteristics | Frequency | Percentage |
|---|---|---|
| **Gender** | | |
| Male | 146 | 56.37 |
| Female | 113 | 43.63 |
| **Age** | | |
| ≤25.00 | 26 | 10.04 |
| 26–35 | 78 | 30.11 |
| 36–45 | 97 | 37.45 |
| ≥45 | 58 | 22.39 |
| **Education** | | |
| Illiterate | 107 | 41.31 |
| Primary | 84 | 32.43 |
| Secondary and Higher Secondary | 45 | 17.37 |
| Bachelors and Above | 23 | 8.88 |
| **Domicile** | | |
| Baluchistan | 74 | 28.57 |
| Punjab | 86 | 33.20 |
| Khyber Pakhtunkhwa | 56 | 21.62 |
| Sindh | 43 | 16.60 |

Source: Survey Data.

Male respondents are greater in number than female respondents. In terms of age, the dominant age group reported during the surveys was 36–45 years. Similarly, the table reveals that majority of the sampled respondents (41.31 percent) were illiterate. Most of the sampled respondents that participated in the surveys (33.20 percent) were from the Punjab province.

Table 2 shows the reliability and validity statistics of the constructs. To determine the reliability of the instrument, its internal consistency was evaluated. Internal consistency, which indicates the degree to which all items in the instrument refer to the same construct, can be assessed by several coefficients, each with their own strengths and limitations [58]. For this study, Cronbach's alpha was calculated using the CFA results.

**Table 2.** Constructs' reliability and validity.

| Construct | Items | Loading | Cronbach's Alfa | AVE |
|---|---|---|---|---|
| Economic Impact | EcoI1 | 0.6872 | 0.8232 | 0.5333 |
| | EcoI2 | 0.6990 | | |
| | EcoI3 | 0.7869 | | |
| | EcoI5 | 0.8107 | | |
| | EcoI6 | 0.7290 | | |
| | EcoI7 | 0.6566 | | |
| Social Impact | SoI1 | 0.9040 | 0.7898 | 0.6937 |
| | SoI2 | 0.7664 | | |
| | SoI3 | 0.8225 | | |
| Cultural Impact | CuI1 | 0.6278 | 0.5892 | 0.5512 |
| | CuI2 | 0.8061 | | |
| | CuI3 | 0.7808 | | |
| Environmental Impact | EnvI1 | 0.6955 | 0.7769 | 0.6881 |
| | EnvI2 | 0.8661 | | |
| | EnvI3 | 0.9112 | | |
| Total Impact | ToI1 | 0.9346 | 0.8368 | 0.8594 |
| | ToI2 | 0.9194 | | |
| | LCS1 | 0.7226 | | |
| Perceived Media Impact | MeI1 | 0.7413 | 0.7227 | 0.6372 |
| | MeI2 | 0.8853 | | |
| | MeI3 | 0.7605 | | |
| Local Community Support | LCS2 | 0.6990 | 0.7478 | 0.5585 |
| | LCS3 | 0.7757 | | |
| | LCS4 | 0.7854 | | |

Source: Authors' Calculations from Survey Data.

Table 2 explains that all the items of the constructs are in the permissible range, as per the instructions of [28]. The cutoff value of the items should be greater than 0.50, hence our instruments depict that there is a sound validity of the variables and they are fit for analysis.

Table 3 represents the cross-loadings of the constructs. All the loadings in their corresponding column show that the loadings are well-placed in their respective columns. This indicates that internal consistency was at an acceptable level.

**Table 3.** Cross-loadings.

| Indicator | Eco | SoI | CuI | EnI | ToI | MeA | LCS |
|---|---|---|---|---|---|---|---|
| EcoI1 | 0.6872 | 0.0494 | 0.7445 | 0.3502 | 0.4398 | 0.3859 | 0.3941 |
| EcoI2 | 0.6990 | 0.1101 | 0.6938 | 0.2935 | 0.4338 | 0.4359 | 0.3969 |
| EcoI3 | 0.7869 | 0.0975 | 0.5299 | 0.4345 | 0.4381 | 0.4724 | 0.4129 |
| EcoI5 | 0.8107 | 0.1343 | 0.5214 | 0.4820 | 0.5231 | 0.5208 | 0.3198 |
| EcoI6 | 0.7290 | 0.0030 | 0.4754 | 0.4083 | 0.4058 | 0.4509 | 0.2569 |
| EcoI7 | 0.6566 | 0.0928 | 0.3275 | 0.4675 | 0.4269 | 0.4102 | 0.2387 |
| SoI2 | 0.1308 | 0.9040 | 0.1040 | 0.0507 | 0.1715 | 0.1400 | 0.0556 |
| SoI3 | 0.0542 | 0.7664 | 0.0331 | 0.0251 | 0.0822 | 0.0687 | 0.0989 |
| SoI4 | 0.0771 | 0.8225 | 0.0574 | 0.0495 | 0.1122 | 0.0236 | 0.0475 |
| CuI1 | 0.3749 | 0.0403 | 0.6278 | 0.2731 | 0.3028 | 0.2924 | 0.3740 |
| CuI2 | 0.6239 | 0.0722 | 0.8061 | 0.2521 | 0.4052 | 0.3579 | 0.3524 |
| CuI3 | 0.6427 | 0.0768 | 0.7808 | 0.2187 | 0.3960 | 0.3970 | 0.3372 |
| EnI1 | 0.3979 | 0.0980 | 0.2546 | 0.6955 | 0.4004 | 0.3364 | 0.2982 |
| EnI2 | 0.4564 | −0.0168 | 0.2536 | 0.8661 | 0.6428 | 0.4472 | 0.2615 |
| EnI3 | 0.5260 | 0.0673 | 0.3114 | 0.9112 | 0.7984 | 0.5138 | 0.2880 |
| ToI1 | 0.5850 | 0.1545 | 0.4727 | 0.7788 | 0.9346 | 0.5989 | 0.3496 |
| ToI2 | 0.5488 | 0.1361 | 0.4532 | 0.6522 | 0.9194 | 0.6403 | 0.3568 |
| MeA1 | 0.5412 | 0.1209 | 0.4596 | 0.3987 | 0.5001 | 0.7413 | 0.2644 |
| MeA2 | 0.5329 | 0.0500 | 0.4080 | 0.4874 | 0.6152 | 0.8853 | 0.3878 |
| MeA3 | 0.3749 | 0.1024 | 0.2413 | 0.3705 | 0.4537 | 0.7605 | 0.2074 |

**Table 3.** *Cont.*

| Indicator | Eco | SoI | CuI | EnI | ToI | MeA | LCS |
|---|---|---|---|---|---|---|---|
| LCS1 | 0.3014 | 0.0409 | 0.2938 | 0.1905 | 0.2128 | 0.2096 | 0.7237 |
| LCS2 | 0.3025 | 0.0664 | 0.2867 | 0.1675 | 0.1849 | 0.2118 | 0.7089 |
| LCS3 | 0.3035 | 0.0433 | 0.3878 | 0.2726 | 0.2998 | 0.2776 | 0.7689 |
| LCS4 | 0.4333 | 0.0687 | 0.4031 | 0.3095 | 0.3739 | 0.3725 | 0.7852 |

Source: Authors' Calculations from Survey Data.

Table 4 expresses the relationships between the constructs through the correlation matrix. All the correlations of the constructs with other constructs are positive and show a significant relationship with one another. Table 4 also disaffirms the chances of multicollinearity, as the independent variables are not very highly correlated to one another. The diagonal values in Table 4 represent the convergent validity, which is within the acceptable range.

**Table 4.** Correlation.

| Construct | 1 | 2 | 3 | 4 | 5 | 6 | 7 |
|---|---|---|---|---|---|---|---|
| Economic Impact | 0.740 | | | | | | |
| Social Impact | 0.194 ** | 0.832 | | | | | |
| Cultural Impact | 0.731 ** | 0.121 | 0.742 | | | | |
| Environmental Impact | 0.534 ** | 0.192 ** | 0.335 ** | 0.829 | | | |
| Total Impact | 0.607 ** | 0.278 ** | 0.492 ** | 0.734 ** | 0.927 | | |
| Media Impact | 0.585 ** | 0.329 ** | 0.456 ** | 0.511 ** | 0.657 ** | 0.798 | |
| Local Community Support | 0.448 ** | 0.125 * | 0.480 ** | 0.313 ** | 0.366 ** | 0.334 ** | 0.746 |

Source: Authors' calculations from Survey Data. ** $p < 0.01$, * $p < 0.05$.

Finally, the environmental impact also has significant value concerning the local community's support for the international project. Our hypotheses H1, H2, H3, H4 are hereby accepted, as their *p*-values are lower than the threshold value. Table 5 also depicts a model summary of the direct predictors and predicted variables. The overall summary shows the explanatory power of the independent variables, as well as the overall significance of the hypothesized model.

**Table 5.** Parameter estimates of the model.

| Model | B | Std. Error | T | Sig. |
|---|---|---|---|---|
| (Constant) | −0.878 | 0.254 | −3.452 | 0.001 |
| Economic Impact | 0.201 | 0.084 | 2.380 | 0.018 |
| Social Impact | 0.155 | 0.052 | 3.003 | 0.003 |
| Cultural Impact | 0.228 | 0.072 | 3.146 | 0.002 |
| Environmental Impact | 0.703 | 0.056 | 12.543 | 0.000 |
| *R*-Square | | 0.63 | | |
| *F*-Value (*p*-Value) | | 108.15 (0.000) | | |

Source: Authors' Calculations from Survey Data.

Table 5 provides the results of the tested hypotheses. The economic impact in terms of the development of the agriculture sector, thereby enhancing livelihoods and income-generating opportunities, has a significant influence on the total impact, which has further affected the local community's support for CPEC. The social factors also have a significant impact on the local populace's support for the CPEC project in Pakistan, approving H5 (see Figure 3). The cultural factors do affect the community support for the CPEC project in the selected areas.

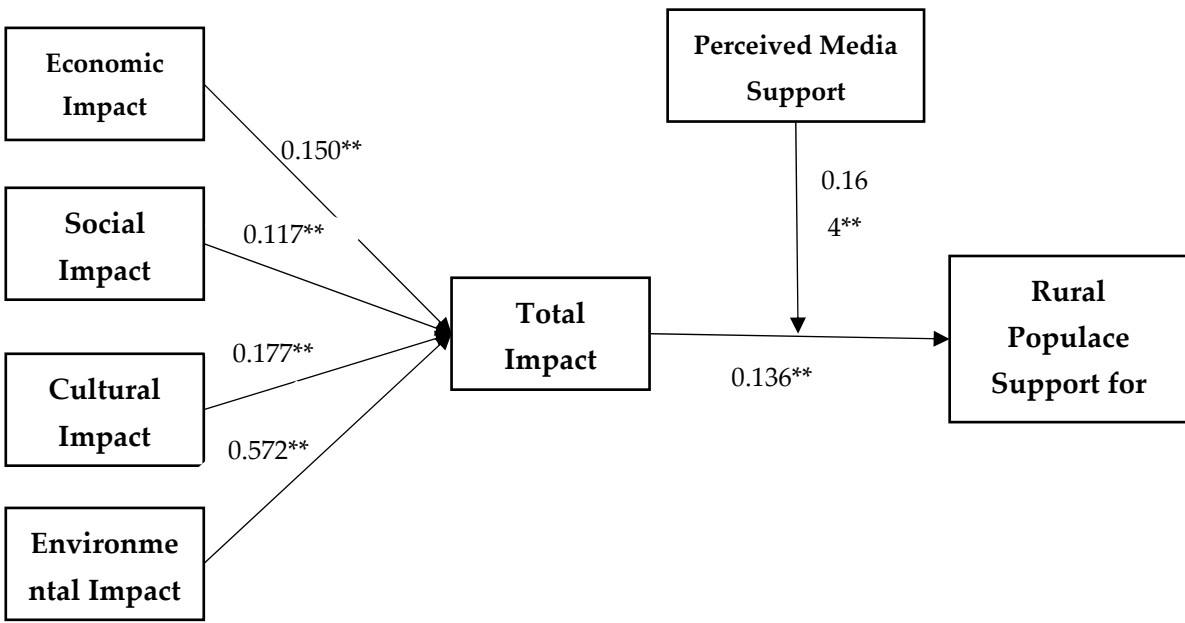

**Figure 3.** Results of the Hypothesized Model. ** *p* < 0.01.

Table 6 represents the model summary with respect to the moderator, i.e., media impact. The moderator shows a 17% change in the relationship between the total impact and local community support for the international project (CPEC). The model gives a significant value, showing that the overall model is significant for the moderation analysis. Table 6 also presents evidence that the moderator (perceived media) influences local community support through the total impact. The media;s influence has been recognized in previous studies to have an immense impact on people's perceptions. This study also testifies that the media's role cannot be overlooked. Our hypothesis 6 (H6) is, therefore, accepted as the coefficient of a moderator that is statistically significant at 5 percent probability level.

**Table 6.** Parameter estimates of the model.

| Model | B | Std. Error | T | Sig. |
|---|---|---|---|---|
| (Constant) | 3.376 | 0.045 | 75.086 | 0.000 |
| Total Impact | 0.136 | 0.054 | 2.501 | 0.013 |
| Media Perceived Support | 0.100 | 0.052 | 1.909 | 0.057 |
| Total Support × Media Perceived Support | 0.087 | 0.034 | 2.572 | 0.011 |
| *R*-Square | | 0.171 | | |
| *F*-Value (*p*-Value) | | 17.48 (0.000) | | |

Source: Authors' Calculations from Survey Data.

## 5. Discussion and Implications

The central aim of the study was to assess the perceptions of the local community in terms of the success and support of the CPEC project in rural areas of Pakistan. Moreover, the study aimed to understand the perceived media influence on the minds of the local community in order to advance the acceptance of CPEC projects. In this regard, the community's perceptions in terms of social, economic, cultural, and environmental factors were considered and probed. Based on primary sources of data collection, the local populace reported perceptions that were personally administered. The study found that the opinions and thinking of the local populace are mainly accepting the project, especially in those rural areas where the CPEC routes are proposed to be constructed. These findings

are consistent with past studies [49,58]. Besides this, the media role was investigated, which has not previously been examined in the context of CPEC's support from local inhabitants.

The implications of the study contribute to the scant research work on the CPEC and the media's role in the perception of the CPEC in rural areas. The social, cultural, economic and environmental impacts were used as predictors of support, and the media as a moderator for the developmental project, which has rarely been assessed before. The support of the local community is undoubtedly crucial for the success of the plans [22]. The result of the current study suggested that, based on the collected data, CPEC will be very significant for the local residents of Pakistan, and will affect their lives socially, culturally, and economically.

Our results on the economic factors signify that the local community in Pakistan thinks that the CPEC will be a harbinger of economic uplift. Our results agree with the previous studies of [15,62], which suggest that the project will give rise to employment, businesses, trade, services, and other commercial activities that will benefit them economically. The societal effects are also positive, showing that the new changes introduced by the international project will be positive and the local people will adjust to the new changes. As was found by Kanwal [22], the project can lead the rural areas of Pakistan to social integrations which will cure the social exclusion issues of Pakistan and China, which, having brotherly relations, will also share some mutually beneficial cultural capital. The cultural impact is also perceived positively by the local community. Moreover, the CPEC is thought to be an addition to the environmental upgrade of the local areas. The project will have a positive impact on the local areas' cleanliness and the green initiatives of the country. Moreover, media influence, a moderator in the study, has a substantial effect on local community perceptions. In concordance with a recent study by Qianqian and Yijun [15], we found that the local media has a massive role in shaping perceptions of the CPEC project in Pakistan. This conclusion is founded on the widespread proliferation of electronic and social media among the local populace. The media influence and tactful handling of its role in the success of the project lie on the shoulders of media regulators and rulers of both countries. Regulators and the government need to utilize the media channel in favor of the project by highlighting its developmental role. The media can also be used as a weapon against the project if not properly handled. All these factors give policy guidelines for the development of the project.

The CPEC is considered as the largest international project, and will make massive contributions not only to the economy of Pakistan, but also to the financial growth of neighboring countries. The project will provide aids and benefits to the local community [1]. The study highlights that the CPEC will be the main source of cooperation among the regional nations, with a notion of shared prosperity and developments. Besides the job creation, economic gains, and business developments, the local people think that the project will help them strategically and politically. Security and insurgency issues, the social disharmony, and national economics vows can be settled by this mega-plan in the long run. As discovered by Kanwal [22], all such benefits could lead the local residents to extend their support to the project for their personal and national interests.

This study has multiple implications. This study is theoretically important for users of social exchange theory in a new cultural context. The more beneficial aspects of the CPEC will keep them in touch with the project, hence the areas may create new developments. The students and teachers with an interest in critical evaluations of international projects may find new implications in the study. CPEC, a buzzword in international cooperation, is also a subject matter for academicians, media personnel, and other various quarters for discussions and analysis. This study may be theoretically helpful for all the concerned quarters. The study, too, has a practical implication on policy formulations and setting the future course of action. The government of Pakistan, the CPEC authority in Pakistan, the Chinese government, and all other relevant bodies may use the findings of this study to obtain further support and accomplish further developments in rural areas of Pakistan. The study is also helpful in furnishing social implications. Past studies also found such

results [63]. Social betterment, societal integration, and social interactions can be strengthened if the findings are considered for implementation. In this regard, social applications can also be of greater importance to make the public aware and gain their support for the mega project.

## 6. Conclusions

Aiming at finding the rural community's support and development for the CPEC project in Pakistan, this study predicts the impact of economic, social, cultural, and environmental effects on community support for the project. The empirical data from 259 members from the local community residing at the routes of the CPEC were used for data analysis. The data were initially cleaned, goodness measures were checked, and then the hypotheses were tested using multiple regression with the latest version of SPSS. Based on the findings, this study concludes that the local community has many economic expectations that the CPEC that will provide better economic opportunities. There are also social imperatives of that the project will add to social development and integration. Culturally, the CPEC will give a positive boost to values, traditions, and other cultural capitals. The study also concludes that the country is in a serious position concerning energy, pollution, and other environmental issues, and that the CPEC will help to overcome these challenges. The media rhas been found to be a major enabler, creating a positive picture of the project, and it can be used to shape the good opinion of the public regarding the CPEC projects. The media impact is undoubtedly crucial for the support of the CPEC in terms changing the rural community's perceptions. The whole study gives an impetus and provides valuable input to policymaking regarding the future success of the CPEC and exploiting its opportunity for Pakistan, and especially for the development of the local rural community of Pakistan.

## 7. Limitations, and Future Research Directions

The CPEC, being a flagship project of the Belt and Road Initiative, is undoubtedly a critical and highly significant venture for both Pakistan and the Chinese government. This study sheds light on several seminal factors that can help to shape the conceptions of the local community to support the project, yet the authors acknowledge several limitations of the study, which could be used as an opportunity for future research studies. The study used convenience sampling; therefore, the findings cannot be used for overall generalization. The findings of the study should be cautiously used by policymakers, keeping in view the context and the usability of the study. Specifically, the data consist of the public opinions of the local community, where their own reporting has been relied upon. The findings related to media impact can be used to improve strategic communication regarding the CPEC and local communities, and to increase public confidence in the Belt and Road Initiative. The media's influence can also be tested in future studies to check solely for the effects of social, print, and electronic media on the success of the CPEC project. This study analyzed the role of media in encouraging support of the CPEC, looking at the public's rating and perceptions. As a next step, archival research and an events study approach to using social media can be applied to obtain more robust and in-depth analysis of the CPEC development. Future research can also use some new variables, like the religious, educational, and health impacts of the CPEC, and their concerns for the project.

**Author Contributions:** Conceptualization, I.U.K. and S.H.; methodology, R.U. and U.G.; software, I.J. and I.U.K.; validation, S.H., I.U.K. and U.G.; investigation, L.J.; resources, L.J.; data curation, S.H.; writing—original draft preparation, I.U.K., A.A., A.R. (Abdul Rauf), A.R. (Abdul Rehman); writing—review and editing, U.G.; visualization, A.A., L.J.; supervision, L.J.; project administration, L.J.; funding acquisition, L.J. All authors have read and agreed to the published version of the manuscript.

**Funding:** This research received no external funding.

**Institutional Review Board Statement:** Not applicable.

**Informed Consent Statement:** Not applicable.

**Data Availability Statement:** All the data are available in the manuscript.

**Acknowledgments:** The author has highly acknowledged the financial support of Chinese Scholarship Council, China.

**Conflicts of Interest:** The authors declare no conflict of interest.

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
