# Peer review of "Development and Sustainability of Rural Economy of Pakistan through Local Community Support for CPEC"

_sustainability, doi:10.3390/su13020686_

Round 1
Reviewer 1 Report
First of all, accept my congratulations on your article.
The study is about to investigate the local community support for CPEC through the lens of Social Exchange Theory, in the rural areas of Pakistan. The study examines the local residents’ perception of CPEC projects in terms of social, economic, cultural, and environmental benefits and costs in order to reveal the local community’s support for that projects. The paper also examines the media influence on local residents. The topics covered by this study are very interesting and important in development processes, because all development must serve the well-being of the people. So, communities perceptions about any initiatives and measures, and their impacts on them are crucial.
The study proved that the surveyed local residents have positive perceptions about the project in question. Thus, it can have a positive impact on the successful implementation of CPEC projects, that in turn can boost the social, economic, cultural, and environmental beneficial impacts on rural communities in Pakistan. The study proved the impact of the media on local residents’ minds.
But, due to the use of convenience sampling, an overall generalization is not possible, which needs to be marked as limitation of the study.
Originality
Research contribution in the paper can be identified, so, the results provide an advance in current knowledge.
Title
The title “Development and Sustainability of Rural Economy of Pakistan Through Local Community’ Support for CPEC” should be corrected as “Development and Sustainability of Rural Economy of Pakistan Through Local Community Support for CPEC”. I suggest that the following be added to the title: “in the light of perceived social, economic, cultural, and environmental impacts”; in accordance with the main aim of the study. Consider it. Anyway, the title reflects the objective and content of the work.
Abstract
The abstract provides a structured overview including aims, and results, conclusion, and implications of key findings.
Introduction
This section introduces the study well, giving an account of the significance, topicality and purpose of the research. The introduction lets the reader know that what authors have written is of interest and relevance to the direct stakeholders and broader interested parties.
Literature Review and Hypotheses
This section is very well based that gives a perfect context for the justification of the research. This section includes many relevant references and authors provide theoretical foundations for the analysis using appropriate references.
Six research hypotheses were formulated in the research. Four of them are related to the local residents’ perception of CPEC projects in terms of social, economic, cultural, and environmental benefits and costs, and the rest are related to perceived total impact and perceived media influence.
It should be noted that the H1-H4 hypotheses need to be reconsidered because of their meanings. So, these hypotheses in their current form are incorrect. Note: something has some or certain effect that can be perceived by someone or e.g. local residents. In this case, CEPEC project or projects has/have effect(s) that the local residents perceive.
Methodology
I propose to divide this section to „Methodology” and „Results”. The section of „Results” can start on page 8. In this section should be provided some more details related to tables and the investigation, the course of the investigation, keeping in mind what is happening and why. Please, give a little more explanation and comments, numbers, percentages, threshold values, etc. Please, indicate the significance level accordingly (** p < 0.01; * p < 0.05).
H1-H4 hypotheses were accepted, but H5 and H6 hypotheses were left out of the evaluation.
“Discussion and Implication”, “Conclusions”, “Limitations, and Future Research Directions”
These sections are okay, they are well based. These sections relate to the analysis and are comprehensive, while contribute to the objectives set out in the introduction section.
The discussion section makes clear the paper’s added value to the existing knowledge and point out the theoretical contributions, which is confirmed by the conclusion. Practical implications, limitations and future research directions are also provided.
References
Please, add DOI numbers to the references
In some places there are misspellings and spelling errors. English proofreading is needed.
Overall, in this point of view the manuscript can be accepted after minor revision.
Recommendation: minor revision
Author Response
Reply to Reviewers Comments for Paper sustainability-1049482
“Development and Sustainability of Rural Economy of Pakistan Through Local Community Support for CPEC”
Dear Editor and Reviewers,
We would like to commence by thanking the editor and the three reviewers for their valuable time and constructive comments. We believe that the comments have been highly constructive and very useful to improve the quality of the revised manuscript. Thus, we have thoughtfully carried out revisions based on the comments and suggestions. Here we provide a point-by-point response to comments and concerns. In addition, all of the changed texts have been highlighted with track changes in our revised manuscript as shown in a word file entitled “revised manuscript with track changes”.
We hope that all these revisions in the revised manuscript will be sufficient to make our manuscript acceptable for publication in Sustainability MDPI. Please let us know if any further clarifications are necessary.
Sincerely,
The Authors
Response for Reviewer #1
- First of all, accept my congratulations on your article.
Response: We are cordially accepted your appreciation.
- The study proved the impact of the media on local residents’ minds. But, due to the use of convenience sampling, an overall generalization is not possible, which needs to be marked as limitation of the study.
Response: Thank you so much for the key suggestion and we have modified our manuscript as per your direction. The changes are highlighted at page number 14, lines 506-507.
- Originality, Research contribution in the paper can be identified, so, the results provide an advance in current knowledge.
Response: Thanks for the good comments. We have highlighted the originality of the manuscript in the conclusion segment. Page No. 14.
- 4. The title “Development and Sustainability of Rural Economy of Pakistan Through Local Community’ Support for CPEC” should be corrected as “Development and Sustainability of Rural Economy of Pakistan Through Local Community Support for CPEC”. I suggest that the following be added to the title: “in the light of perceived social, economic, cultural, and environmental impacts”; in accordance with the main aim of the study. Consider it. Anyway, the title reflects the objective and content of the work.
Response: We are obliged for the good suggestion and we have modified the title in our revised manuscript “Development and Sustainability of Rural Economy of Pakistan Through Local Community Support for CPEC”.
- 5. The abstract provides a structured overview including aims, and results, conclusion, and implications of key findings.
Response: Thank you so much for the critical review, we have modified the abstract accordingly in our revised manuscript.
- 6. This section introduces the study well, giving an account of the significance, topicality and purpose of the research. The introduction lets the reader know that what authors have written is of interest and relevance to the direct stakeholders and broader interested parties.
Response: We are much obliged. We have made some necessary deletion and addition in the introduction segment so it may highlight the clear background of the study.
- Literature Review and Hypotheses; This section is very well based that gives a perfect context for the justification of the research. This section includes many relevant references and authors provide theoretical foundations for the analysis using appropriate references.
Response: Thanks for the appreciation.
- Six research hypotheses were formulated in the research. Four of them are related to the local residents’ perception of CPEC projects in terms of social, economic, cultural, and environmental benefits and costs, and the rest are related to perceived total impact and perceived media influence.
It should be noted that the H1-H4 hypotheses need to be reconsidered because of their meanings. So, these hypotheses in their current form are incorrect. Note: something has some or certain effect that can be perceived by someone or e.g. local residents. In this case, CEPEC project or projects has/have effect(s) that the local residents perceive.
Response: Thank you so much for the constructive comments. And we have revised the Hypothesis as per your suggestion. And highlighted as below
H1. The perceived economic impact of local residents has a positive effect on the local residents’ support for the CPEC projects.
H2. The perceived social impact of local residents has a positive effect on the local residents’ support for the CPEC projects.
H3. The perceived cultural impact of local residents has a positive effect on local residents’ support for the CPEC projects
H4. The perceived environmental impact of local residents has a positive effect on the local residents’ support for the CPEC projects.
- Methodology; I propose to divide this section to „Methodology” and „Results”. The section of „Results” can start on page 8. In this section should be provided some more details related to tables and the investigation, the course of the investigation, keeping in mind what is happening and why. Please, give a little more explanation and comments, numbers, percentages, threshold values, etc. Please, indicate the significance level accordingly (** p < 0.01; * p < 0.05).
Response: We acknowledge the key comments. We have separated the methodology and results in our revised manuscript accordingly.
- “Discussion and Implication”, “Conclusions”, “Limitations, and Future Research Directions” These sections are okay, they are well based. These sections relate to the analysis and are comprehensive, while contribute to the objectives set out in the introduction section. The discussion section makes clear the paper’s added value to the existing knowledge and point out the theoretical contributions, which is confirmed by the conclusion. Practical implications, limitations and future research directions are also provided.
Response: Thanks and your agreement are highly appreciated.
- In some places there are misspellings and spelling errors. English proofreading is needed.
Response: We have throughly checked the grammatical and spelling errors in our revised manuscript.

Reviewer 2 Report
Presentation of an interesting research, but it needs an overall review to deal with expression details, at least.
Also in terms of organization, we could find only in lines 407-408 the information: "Quantitative data analysis approach through SPSS was applied to data collected from the residents of seven districts (...)", while it was supposed to have presented this before, e.g. within "3.4 Data Analysis"...
Adjectives are not always adequate to academic standards. Just some few examples:
32. The article is "exuberantly important"
42-43. "The friendship (...) is considered as deep as the ocean"
254-255. "its all-weather friend Pakistan".
274. "perennial growth of the Chinese economy" (continuing?)
471. "golden opportunity for Pakistan"
There are some information gaps or misspellings. We present only some, as examples, but much more small failures in the English expression (including some syntax) are found. Mere examples follow, after a quick reading. Other sentences should be reviewed, too:
45-46. "During his visit, the Chinese premier, Li Keqiang, worked for launching the mutual project for the socio-economic and infrastructural up-gradation of Pakistan, called CPEC" (readers don't know the date - the year, at least - of the visit)
115. "the Chines government" (Chinese?)
217. "the local papulations"
274. "(...) the Chinese economy and the success stories of their (...)"
341. Age... ≤25.00
400. "These findings correspond to the findings of previous studies These findings are (...)"
456-457. "(...) this study predicts the economic, social, cultural, and environmental effects on the community support total support for the project." (sentence to reformulate...)
461-462. "(...) the CPEC that it will change their economic miseries and will give them respectable employment". (sentence to reformulate...)
465-466. "(...) the CPEC will heal and promote the environmental worries of Pakistan." (sentence to rethink / reformulate...)
Author Response
Reply to Reviewers Comments for Paper sustainability-1049482
“Development and Sustainability of Rural Economy of Pakistan Through Local Community Support for CPEC”
Dear Editor and Reviewers,
We would like to commence by thanking the editor and the three reviewers for their valuable time and constructive comments. We believe that the comments have been highly constructive and very useful to improve the quality of the revised manuscript. Thus, we have thoughtfully carried out revisions based on the comments and suggestions. Here we provide a point-by-point response to comments and concerns. In addition, all of the changed texts have been highlighted with track changes in our revised manuscript as shown in a word file entitled “revised manuscript with track changes”.
We hope that all these revisions in the revised manuscript will be sufficient to make our manuscript acceptable for publication in Sustainability MDPI. Please let us know if any further clarifications are necessary.
Sincerely,
The Authors
Response for Reviewer #2
- Presentation of an interesting research, but it needs an overall review to deal with expression details, at least.
Response: Thanks so much for the constructive comments. The paper has been critically revised as per your suggestion.
Also in terms of organization, we could find only in lines 407-408 the information: "Quantitative data analysis approach through SPSS was applied to data collected from the residents of seven districts (...)", while it was supposed to have presented this before, e.g. within "3.4 Data Analysis"...
Response: Many thanks to realise the key point. We highlighted “Quantitative data analysis approach through SPSS was applied to data collected from the residents of seven districts lying on the CPEC route in the Baluchistan, Punjab, and KP, Pakistan” in our revised manuscript. At page No. 8, lines 350-353.
- Adjectives are not always adequate to academic standards. Just some few examples:
- The article is "exuberantly important"
42-43. "The friendship (...) is considered as deep as the ocean"
254-255. "its all-weather friend Pakistan".
- “perennial growth of the Chinese economy” ( continuing ?)
- "golden opportunity for Pakistan"
Response: Thanks for the productive comment. We have changed the un-adequate adjectives in our revised manuscript and the change highlighted at page no. 1, 6 and 14.
- There are some information gaps or misspellings. We present only some, as examples, but much more small failures in the English expression (including some syntax) are found. Mere examples follow, after a quick reading. Other sentences should be reviewed, too:
Response: Thank you so much for the good comments. We have followed your suggestion accordingly in our revised draft.
45-46. "During his visit, the Chinese premier, Li Keqiang, worked for launching the mutual project for the socio-economic and infrastructural up-gradation of Pakistan, called CPEC" (readers don't know the date - the year, at least - of the visit)
Response: Thanks for the identification of key point. We have excluded this paragraph from the revised manuscript. And highlighted at page 1 and lines 44-45.
LIN 203: There are other more recent papers than that by Card et al. 2013, for example: Fraser et al. Theriogenology 2020; Mankowska A et al. International Journal of Molecular Science 2020.
115."the Chines government" (Chinese?)
- "the local papulations"
- "(...) the Chinese economy and the success stories of their (...)"
- Age... ≤25.00
Response: Thank you so much and we have fixed the grammatical errors in our revised manuscript.
400. "These findings correspond to the findings of previous studies These findings are (...)"
Response: Thank you so much for pointing the deficiency. We have been re-structured the sentence in the revised manuscript. Page no. 12, lines 409-410.
456-457. "(...) this study predicts the economic, social, cultural, and environmental effects on the community support total support for the project." (sentence to reformulate...)
Response: As per your kind suggestion, we have rephrased the paragraph in revised draft. Highlighted at page 14, lines 485-500.
- 461-462. "(...) the CPEC that it will change their economic miseries and will give them respectable employment". (sentence to reformulate...)
Response: We have been reformulated the sentence in the revised draft.
- 465-466. "(...) the CPEC will heal and promote the environmental worries of Pakistan." (sentence to rethink / reformulate...)
Response: We have been reconsidered the sentence in the revised draft.

Reviewer 3 Report
The article is definitely a significant contribution based on a large-scale survey among rural residents. The methodology is sound and well substantiated the results are briefly but sufficiently reflected. The only thing I could suggest are some graphic visualisations of the results, as well as possibly a map of the discussed region, as to a foreigner the peculiarities are not clear enough. But this is up to the authors to decide, the article is already a well-prepared contribution.
Author Response
Reply to Reviewers Comments for Paper sustainability-1049482
“Development and Sustainability of Rural Economy of Pakistan Through Local Community Support for CPEC”
Dear Editor and Reviewers,
We would like to commence by thanking the editor and the three reviewers for their valuable time and constructive comments. We believe that the comments have been highly constructive and very useful to improve the quality of the revised manuscript. Thus, we have thoughtfully carried out revisions based on the comments and suggestions. Here we provide a point-by-point response to comments and concerns. In addition, all of the changed texts have been highlighted with track changes in our revised manuscript as shown in a word file entitled “revised manuscript with track changes”.
We hope that all these revisions in the revised manuscript will be sufficient to make our manuscript acceptable for publication in Sustainability MDPI. Please let us know if any further clarifications are necessary.
Sincerely,
The Authors
Response for Reviewer #3
- The article is definitely a significant contribution based on a large-scale survey among rural residents. The methodology is sound and well substantiated the results are briefly but sufficiently reflected. The only thing I could suggest are some graphic visualisations of the results, as well as possibly a map of the discussed region, as to a foreigner the peculiarities are not clear enough. But this is up to the authors to decide, the article is already a well-prepared contribution.
Response: Thank you so much for the agreement towards our study.
